# Liver and Pancreatic Involvement in Children with Multisystem Inflammatory Syndrome Related to SARS-CoV-2: A Monocentric Study

**DOI:** 10.3390/children9040575

**Published:** 2022-04-18

**Authors:** Antonietta Giannattasio, Marco Maglione, Carolina D’Anna, Stefania Muzzica, Serena Pappacoda, Selvaggia Lenta, Onorina Di Mita, Giusy Ranucci, Claudia Mandato, Vincenzo Tipo

**Affiliations:** 1Pediatric Emergency Unit, Santobono-Pausilipon Children’s Hospital, AORN, 80130 Naples, Italy; marcomaglione84@gmail.com (M.M.); dannacarol@alice.it (C.D.); stefaniamuzzica@hotmail.it (S.M.); serena.pappacoda@gmail.com (S.P.); selvylenta@hotmail.com (S.L.); onorina8@libero.it (O.D.M.); giusyranucci85@gmail.com (G.R.); enzotipo@libero.it (V.T.); 2Medical Department, University of Salerno, 84084 Salerno, Italy; cla.mandato@gmail.com

**Keywords:** multisystem inflammatory syndrome related to SARS-CoV-2, children, acute liver injury, pancreatic injury, prognosis

## Abstract

Liver and pancreatic involvement in children with Multisystem Inflammatory Syndrome related to SARS-CoV-2 (MIS-C) has been poorly investigated so far. We reviewed a cohort of MIS-C patients to analyze the prevalence of acute liver injury (ALI) and pancreatic injury and their correlation with clinical outcomes. Demographic, clinical, laboratory and imaging features of children with MIS-C at admission and during hospital stay were prospectively collected. Fifty-five patients (mean age 6.5 ± 3.7 years) were included. At admission, 16 patients showed ALI and 5 had increased total serum lipase. During observation, 10 more patients developed ALI and 19 more subjects presented raised pancreatic enzymes. In comparison to those with normal ALT, subjects with ALI were significantly older (*p* = 0.0004), whereas pancreatic involvement was associated to a longer duration of hospital stay compared with patients with normal pancreatic enzymes (*p* = 0.004). Time between hospital admission and onset of ALI was shorter compared to the onset of raised pancreatic enzymes (3.2 ± 3.9 versus 5.3 ± 2.7 days, respectively; *p* = 0.035). Abdominal ultrasound showed liver steatosis in 3/26 (12%) and hepatomegaly in 6/26 (16%) patients with ALI; 2 patients presented enlarged pancreas. Although liver and pancreatic involvement is commonly observed in MIS-C patients, it is mild in most cases with a complete recovery.

## 1. Introduction

Since early 2020, evolution of the SARS-CoV-2 pandemic has shown a progressively increasing involvement of children and, consequently, of pediatricians. Initially regarded as the age group less frequently and less severely affected by the infection, children have soon shown to be not only a potential target for SARS-CoV-2, generally with mild clinical manifestations, but, more importantly, have demonstrated to be at risk of more severe post-infectious complications, mainly represented by the Multisystem Inflammatory Syndrome related to SARS-CoV-2 (MIS-C) [1,2]. This condition, which has clinical overlap with Kawasaki disease, toxic shock syndrome and macrophage activation syndrome, presents with extreme inflammation and multiorgan involvement and is believed to result from an abnormal immune response to the virus [3]. In both acute SARS-CoV-2 infection and MIS-C, gastrointestinal manifestations may represent a relevant aspect of the clinical picture. In acute SARS-CoV-2 infected children, diarrhea, vomiting and abdominal pain represent the most frequent gastrointestinal manifestations [4,5]. Less typically, acute liver injury (ALI), defined as an elevation in aminotransferases with or without cholestasis, has been reported in a wide range of children with COVID-19 [6,7,8,9] but with a substantial agreement on the mild nature of the virus-associated hepatic involvement. Data on pancreatic involvement during SARS-CoV-2 infection are sparser. Few reports of acute pancreatitis during acute infection have been published and generally described a mild clinical course with no complications [10,11].

With regards to MIS-C, while ALI is a common finding in patients presenting with shock and/or multi-organ failure [12], the role and prognostic significance of mild hepatitis in not severe cases are still debated [13,14]. Even less data on pancreatic involvement in MIS-C are available. 

Aims of the present study were: (1) to investigate the prevalence of liver and pancreatic involvement in a cohort of children with MIS-C; (2) to evaluate clinical, laboratory and imaging profile of these patients and to correlate it with disease’s outcomes.

## 2. Materials and Methods

### 2.1. Study Design and Subjects

All patients up to 14 years of age consecutively hospitalized at the “Santobono-Pausilipon” Children’s Hospital with a diagnosis of MIS-C between September 2020 and September 2021 were eligible for the enrollment. Diagnosis of MIS-C was performed according to the CDC criteria [1]. Data on demographics, exposure history, medical history, clinical presentation, laboratory tests, and imaging studies were collected. Patients were prospectively enrolled, and only those with a definitive diagnosis of MIS-C and complete medical chart reviews were included in the study. Patients suspected of being affected by MIS-C at presentation whose diagnosis was not confirmed during hospital stay were not recruited. Subjects referred to our hospital after diagnostic and therapeutic management was started elsewhere were excluded if a complete laboratory assessment at presentation was not available. Obesity was defined as body mass index (BMI) >95th percentile for age and sex. 

### 2.2. Procedures

Laboratory investigations at admission included complete blood count, inflammatory markers (C-reactive protein, CRP, procalcitonin, PCT, ferritin, interleukin 6, IL6), kidney and liver functional tests, electrolytes, albumin, troponin, b-type natriuretic peptide (BNP), SARS-CoV-2 serologic assay and naso-pharyngeal swab for SARS-CoV-2 detection. Tests aimed at excluding infections from other etiologies were performed in all children (blood-, stool- and urine-cultures; multiplex nucleic acid amplification test for multiple respiratory pathogens; Epstein–Barr virus and cytomegalovirus antibodies). Peripheral blood smear was also performed in all cases. All patients underwent abdominal ultrasound and echocardiography. 

Liver involvement assessment included: markers of hepatocellular injury (aspartate aminotransferase, AST, alanine aminotransferase and ALT), markers of cholestasis (gamma glutamyl transferase, GGT, alkaline leukocyte phosphatase, ALP and bilirubin) and of synthetic liver function (albumin and total proteins) and hemostasis (INR and prothrombin time). ALI was defined in presence of ALT elevation > 40 U/L. ALI was defined as severe in case of ALT >200 U/L. In presence of hypertransaminasemia, other causes of liver disease were excluded (namely, major and minor hepatotropic viral infection, urinary tract infection, metabolic disorders and drug induced liver injury).

Pancreatic involvement was defined based on the values of amylasemia (AMS) and lipasemia. Hyper-AMS and hyper-lipasemia were defined for values over the upper limit of normality (ULN, <80 U/L and <60 U/L for AMS and lipase, respectively). Acute pancreatitis was diagnosed according to the Atlanta Criteria [15,16].

Unless otherwise indicated based on the clinical situation, laboratory tests were collected on a daily basis for the first 3 days after admission and every 24–72 h for the following days, according to general condition and severity of disease, until discharge. Additional laboratory monitoring was performed in the outpatient setting with new laboratory tests approximately 7 days after discharge (or last in-hospital assessment), at the time of steroid therapy discontinuation and again after one month. The following laboratory assessments were performed every 3 to 6 months based on initial clinical impairment and response to treatment for a period of 12 to 18 months after discharge.

Abdominal ultrasound was performed both at admission and at discharge and for patients showing ALI or raised pancreatic enzymes repeated at the time of detection of abnormal liver and/or pancreatic tests. The same experienced pediatric radiologist performed all the imaging studies. Ultrasound studies were repeated approximately one month after discharge in all patients showing ALI, pancreatic involvement or previously detected imaging abnormalities.

Treatment strategies were supported by the American College of Rheumatology Clinical Guidance and by the treatment guidance from the Rheumatology Study Group of the Italian Society of Pediatrics [17,18]. According to these guidelines, treatment of MIS-C was supportive along with IV immunoglobulins (IVIG, 2 g/kg in 12–24 h) alone or combined with high (10–30 mg/kg) or low (2 mg/kg) doses of methylprednisolone, depending on the severity of the condition. Biological drugs, namely anakinra, were added in case of failure of standard treatments. 

### 2.3. Statistical Analysis

Means and standard deviation (SD) for continuous variables, and numbers and percentages for categorical variables, were used. Demographic, clinical and laboratory features were compared between patients with and without ALI and between patients with and without pancreatic involvement. Categorical variables were evaluated with chi-square test and Fisher’s test, as appropriate. Wilcoxon’s rank sum test was used for continuous variables. A two tailed *p* value < 0.05 was considered statistically significant. Analysis was performed on StataCorp LLC Stata 13.0 (College Station, TX, USA).

## 3. Results

### 3.1. Enrolled MIS-C Population

A total of 55 children with diagnosis of MIS-C were enrolled. Clinical data are summarized in Table 1. All but 2 patients (a female from South-East Asia and a male from Nigeria) were of Caucasian ethnicity. Only one patient had a comorbidity (type 1 diabetes). No patient had a history of chronic liver or pancreatic diseases. Except for fever, present in all cases, conjunctivitis and skin rash were the most frequent single signs at presentation. Overall, abdominal symptoms (pain and/or vomiting and/or diarrhea) were present in 45 (82%) patients at hospital admission. As for cardiac injury, 27 (49%) patients had high troponin level, and 29 (53%) had abnormalities at cardiac ultrasound imaging. Five (9%) patients received IVIG alone; 27 (49%) patients received IVIG + high-dose steroids, and 23 (42%) received IVIG + low-dose steroids. In 8 (15%) cases with poor or uncomplete response to first-line therapy, anakinra was used in addition to other treatments with rapid clinical and laboratory improvements. In one child with type 1 diabetes and a severe MIS-C with cardiac dysfunction, anakinra was used as first line therapy in addition to IVIG and low doses of steroids because of the risk of a poor glycemic control and severe hyperglycemia due to intravenous boluses of steroids.

No patient required oxygen supply or invasive ventilation. 

### 3.2. Liver Involvement

Overall, 26 (47%) MIS-C patients showed ALI (mean AST value 94 ± 67 IU/L; mean ALT value 104 ± 84 IU/L) during observation, with signs of mild cholestasis in 2 cases. Of these, 16 (29%) patients already had ALI (mean AST value 106 ± 83 IU/L; mean ALT value 122 ± 105 IU/L) at hospital admission and 10 (18%) additional cases developed ALI (mean AST value 47 ± 29 IU/L; mean ALT value 85 ± 45 IU/L) during hospital stay. The mean highest ALT value during hospital observation was 133 ± 102 IU/L. No patient developed acute liver failure. All the subjects presenting with ALI at admission had a mild increase in liver enzymes, with only one child presenting with severe hypertransaminasemia (AST 307 IU/L, ALT 480 IU/L). This was an otherwise healthy 5-year-old girl presenting with a severe myocardial dysfunction requiring intensive care unit admission. She received treatment with IVIG + high doses of steroids + anakinra. The patient obtained a full recovery of cardiac damage and normalization of transaminases within 7 days after hospital discharge (length of hospital stay: 16 days). Onset of ALI in the additional 10 patients was observed after a mean hospital stay of 7 (± 3) days, and 3 children out of 10 (30%) developed a severe ALI. Despite not statistically significant, the subgroup of children developing ALI during hospital stay showed a trend towards shorter duration of symptoms before admission in comparison to the subgroup presenting with ALI at admission (3.7 ± 1.4 versus 4.4 ± 2.2 days, *p* = 0.08). Fifteen (58%) patients out of 26 still showed hypertransaminasemia at discharge (mean ALT value 65 ± 47 IU/L). For these subjects, time to complete normalization of laboratory values was 9 ± 5 days. Table 2 summarizes laboratory data and clinical parameters of children with and without ALI. A raise in liver enzymes was more frequently found in males, and children with ALI were significantly older than the not-ALI group. Although all inflammatory parameters were higher in the ALI-group compared with not-ALI patients, the difference did not reach a statistical significance (Table 2). However, when patients were divided into 2 subgroups according to ferritin levels (> or ≤1000 ng/mL), we found that the percentage of patients with ALI had more frequently ferritin above this cut-off compared to patients in the not-ALI group (8/26, 31%, versus 2/29, 7%; *p* = 0.02). This difference was confirmed also when a cut-off of 680 ng/mL was used, according to the criteria for the definition of macrophage activation syndrome (ferritin >680 ng/mL in 11/26 patients, 42%, in the ALI group versus 5/29 patients, 17%, in not-ALI group, *p* = 0.04). No significant correlation with other inflammatory parameters (CRP and IL6 levels) was found. As for cardiac imaging, it is to note that 5/26 (19%) patients with ALI had myocarditis versus 1/29 (3%) in the not-ALI group, although the difference did not reach a statistical significance (*p* = 0.06). As well, no difference was found between the 2 groups for other cardiac findings as pericardial effusion and left ventricle dysfunction. 

With regards to liver imaging, 3 out of 26 (12%) patients with ALI showed a bright liver at ultrasound and 6 (23%) subjects showed hepatomegaly, with both findings being present in 3 cases. Liver ultrasound was unremarkable in the remaining 17 (65%) patients. Steatosis was severe only in one obese patient (male, 12 years) who showed the same imaging features at the liver ultrasound performed after 30 days. In the remaining 2 cases, steatosis was mild and disappeared at the post-discharge abdominal ultrasound. During follow-up, hepatomegaly disappeared in all cases. In the not-ALI group, one patient had hepatomegaly, and one child had a mild steatosis (no significative difference with the ALI-group). The latter was an obese 4-year-old girl who showed persistent steatosis at a liver ultrasound performed after 14 days. Type of treatment did not significantly differ between ALI and not-ALI groups (data not shown). 

### 3.3. Pancreatic Involvement

Overall, raised pancreatic enzymes were found in 24 (14%) patients, with a mean AMS value of 180 ± 64 U/L and a mean lipase value of 191 ± 135 U/L. Five children presented pancreatic injury at hospital admission (mean AMS value 175 ± 73 U/L; mean lipase value 127 ± 57 U/L), while additional 19 patients developed pancreatic disease during hospital stay (mean AMS value 137 ± 62 U/L; mean lipase value 208 ± 142 U/L). Mean highest AMS and lipase values during hospital observation were 181 ± 116 and 252 ± 128 U/L, respectively. No patient fulfilled the criteria for acute pancreatitis. Other causes of raised AMS and lipase (such as acute cholecystitis, gallstones and bowel obstruction) were excluded. Seven (29%) patients still presented raised pancreatic enzymes at discharge (mean AMS value 101 ± 58 U/L, mean lipase value 152 ± 193 U/L). Normalization of laboratory values was observed 21 ± 8 days after hospital discharge. Laboratory features in the 2 groups of patients (with and without pancreatic involvement) are reported in Table 3. Unlike ALI, patients with pancreatic involvement had a more uniform distribution with regards to age and gender, but a longer hospital stay compared to patients with normal pancreatic enzymes. Abdominal pain was one of the presenting symptoms in both the 17 (71%) patients of the group with pancreatic involvement and the 17 (55%) patients with normal pancreatic enzymes (*p* = ns). Inflammatory parameters did not differ between the two groups. Unlike ALI, the proportion of patients with ferritin >680 ng/mL was similar between the group with raised pancreatic enzymes (4/24, 17%) and patients with no pancreatic injury (8/31, 26%). As in ALI patients, troponin was more frequently increased in patients with pancreatic involvement (16/24, 67%) compared to patients with no raise of pancreatic enzymes (11/31, 35%) (*p* = 0.02). Mean BNP did not differ between the two groups (438 ± 604 pg/mL versus 336 ± 530 pg/mL in presence or absence of pancreatic involvement, respectively). 

Complete recovery of pancreatic abnormalities was obtained in all patients with pancreatic injury. 

With regards to pancreatic imaging, only 2 (4%) subjects presented mild pancreas enlargement (one with and the other without pancreatic involvement). In both cases, pancreatic imaging was unremarkable at the follow-up ultrasound.

Again, type of treatment did not significantly differ between the 2 groups (data not shown). 

It is of note that the mean period to the detection of raised pancreatic enzymes was longer compared to the time for the onset of ALI (5.3 ± 2.7 days versus 3.2 ± 3.9 days; *p* = 0.035).

## 4. Discussion

The present study provides an assessment of liver and pancreatic involvement in a moderately sized cohort of children with MIS-C admitted to the main pediatric tertiary care center of Southern Italy. We analyzed not only baseline data but also the development of liver and pancreatic involvement during hospital observation. Our main finding was that, even though generally mild, involvement of these two organs is frequent, affecting almost half of patients. These findings are only partially in line with previous reports. 

While an acute COVID-19 liver involvement seems to affect mainly younger children (<3 years of age) [19], we found a higher prevalence of ALI in older MIS-C children. This is in line with the work by Perez et al. who found that liver involvement is more frequent in older MIS-C patients [9]. It has been hypothesized that the mechanism of liver injury in acute SARS-CoV-2 infection and in MIS-C may be different. In COVID-19, liver damage could be related, at least in part, to a direct cytopathic effect of the virus given the higher expression of angiotensin-converting enzyme 2 (ACE2) receptors (which are considered the cellular receptor for SARS-CoV-2) on the surface of the liver and bile duct epithelial cells [20,21]. In MIS-C, an immune-mediated pathogenetic mechanism has been postulated [14]. The massive release of pro-inflammatory cytokines observed in MIS-C is accompanied by organ dysfunction, including liver damage, similarly to what was observed in adults with severe COVID-19 [22]. However, while in acute COVID-19 IL6 has been specifically implicated in determining liver damage [23], in our study, IL6 levels did not differ between patients with or without ALI. On the other hand, we found an association of ALI with high levels of ferritin. Despite the absence of similar relationships with other inflammatory markers, particularly CRP and IL6, should this finding be confirmed in larger cohorts of patients, the direct association between ALI and systemic inflammation would be supported. 

Within the ALI group, about 20% of patients had normal liver tests at admission and developed ALI during hospital observation. This finding is cumbersome to interpret, but a hypothesis could be the non-statistically significant trend towards shorter duration of symptoms before admission. Whether confirmed on larger populations, this finding would represent the most likely explanation. Furthermore, the absence of significant differences in the clinical course or response to treatment suggests that these patients are phenotypically similar to those with ALI at presentation. 

Patients with ALI had higher troponin levels at baseline compared to not-ALI patients. Conversely, BNP did not differ between the two groups. Cardiac markers, including BNP and troponin, are used to predict the progression of deterioration in earlier phases of MIS-C [24]. However, the significance of their elevation is uncertain and should not necessarily trigger an evaluation or treatment for heart failure unless there is clear clinical evidence for the diagnosis. BNP is primarily synthesized and released from the ventricle in response to ventricular hemodynamic changes and stress. Although increased levels of BNP have been shown to predict possible death and cardiovascular diseases in people without heart failure, causes of elevated BNP do not necessarily have to be related to a heart dysfunction. In MIS-C patients, there is water retention with serositis (pleural and pericardial effusion and free fluid in the abdomen) [25] which can lead to an elevation of BNP. Troponin elevation is more strictly related to cardiac damage than BNP in MIS-C. In our study, a significative higher number of patients in the ALI group had myocarditis compared to the not-ALI group. We can hypothesize that troponin and not BNP is higher in the ALI group compared with not-ALI patients because of a higher rate of real cardiac dysfunction (see rate of myocarditis) in the first group compared to the not-ALI group.

As for the prognostic significance of ALI in MIS-C, no definitive/clear data are available to date. Perez and coworkers reported an association with a more severe course of the disease and with longer hospitalization [9]. Conversely, Lazova et al. did not find an association between the severity of liver injury and adverse outcome in 19 MIS-C patients aged 1 to 17 years [14]. Our results in a larger sample confirmed that the presence of ALI did not represent an unfavorable prognostic factor for medium term clinical outcomes. 

Prevalence of pancreatitis in MIS-C varies from 3% to more than half of cases [26,27]. Acharyya et al. reported a very high rate (53%) of acute pancreatitis in a small cohort of 17 MIS-C patients [27]. On the other hand, sporadic cases of acute pancreatitis in MIS-C have been reported [28,29]. A recent study analyzed the fraction of children with MIS-C presenting as acute pancreatitis and found that 9% of patients had pancreatitis since the onset of the disease [27].

Our data showed a lower prevalence of pancreatic involvement in MIS-C. Furthermore, even though almost all our patients presented with abdominal pain, in no case this symptom was related to an ongoing acute pancreatitis. Interestingly, while in MIS-C alteration of liver enzymes is often detected at admission, pancreatic enzymes levels usually raised later during the hospital stay once the treatment had been started. This finding has not been previously described. Furthermore, pancreatic injury did not correlate with the severity of MIS-C or with the type of therapy.

The reason for the elevation of pancreatic enzymes in MIS-C patients who did not fulfill criteria for acute pancreatitis is unclear. Capillary leakage due to obstruction of venous and lymphatic drainage of pancreatic and peripancreatic tissues due to mesenteric inflammation may be one of the involved mechanisms. Furthermore, transperitoneal absorption of amylase due to inflammation-related increased peritoneal permeability may represent another likely cause [30]. In addition, patients with MIS-C often have diarrhea, which might cause elevated serum AMS and lipase. Indeed, it is known that diarrhea or intestinal inflammation increase the absorption of amylase and lipase in the bowel lumen, with further absorption into the blood [31,32]. Furthermore, AMS is cleared by the reticuloendothelial system [32,33]. Therefore, as in MIS-C a disarrangement of immune cell profiling has been hypothesized, we can assume that, in addition to the above-mentioned explanations, the detection of high pancreatic enzymes may also be related to an impairment of the reticuloendothelial system. Other studies have reported hyper-AMS in adults with COVID-19 [34]. In this case, renal impairment has been reported to be one contributing factor to the elevated serum AMS value. In the present study, none of our patients had acute kidney injury at onset nor developed it during hospital observation.

Although ALI and increased pancreatic enzymes do not seem to have a relevant clinical impact in most cases, whether these alterations affect patients’ outcomes is still unclear. Our analysis showed no correlation with inflammatory markers typically raised in MIS-C patients, thus suggesting that liver and pancreatic involvement does not represent a marker of more severe multisystem inflammation. Nevertheless, patients with pancreatic involvement faced a significantly longer hospitalization in comparison to those without pancreatic involvement. The reasons for this finding may be several. Given the absence of correlation with the inflammatory markers, the hypothesis that pancreatic involvement is associated with more severe or prolonged symptoms entailing the need for a longer hospital stay seems unlikely. More probably, raised pancreatic enzymes may induce clinicians to a more cautious approach, with repeated determinations before patient’s discharge, in order to rule out acute pancreatitis or, at least, worsening of pancreatic involvement. Interestingly, this is not observed in case of hyper-ALT, likely because transaminases are typically managed with less frequent and more spaced follow-up controls.

Our study has some limitations. First, as the study was monocentric, our population was limited, and some of our findings that did not achieve statistical significance needed larger populations to be generalized. Moreover, as patients were enrolled in a tertiary care pediatric hospital, which is charged with the management of children younger than 14 years, subjects beyond this age were not included. As ALI was more frequently found in older children, inclusion of adolescents would have been interesting in order to explore whether this finding is confirmed in patients aged more than 14 years. Furthermore, it is possible that we have missed some correlation because of the small size of the single subgroups. 

In conclusion, liver and pancreatic involvement were commonly observed in a moderately sized cohort of MIS-C patients. However, the organ impairment was not severe in the majority of cases, and it did not seem to be correlated with a poor prognosis. Should these results be confirmed in larger cohorts, it is likely that excessively frequent liver and pancreatic function testing will result unnecessary, particularly if more robust data suggest that this practice is not outcome-altering. Nevertheless, given the areas of uncertainty still to be clarified, we suggest including complete liver and pancreatic assessments in the current evaluation of MIS-C since its onset. Furthermore, patients with MIS-C also require careful monitoring throughout hospital stay in order to promptly detect liver and pancreatic injury and to identify those patients who develop an acute pancreatitis. Further studies on larger samples of MIS-C patients are needed to provide insights into the pathophysiology of hypertransaminasemia and increased pancreatic enzymes in this condition. 

## Figures and Tables

**Table 1 children-09-00575-t001:** Clinical and laboratory findings in 55 patients with MIS-C at hospital admission.

Variable	*n* (%)
Males ^+^	29 (53)
Age (years) *	6.5 ± 3.7
Obesity ^+^	8 (14.5)
Duration of symptoms before hospital admission (days) *	3.96 ± 2.12
Clinical presentation ^+^	
- Fever	55 (100)
- Conjunctivitis	37 (67)
- Rash	30 (54)
- Abdominal pain	31 (56)
- Diarrhea	28 (51)
- Meningism	9 (16)
- Chest pain	3 (5)
No. of patients with ALI ^+^	16 (29)
No. of patients with raised pancreatic enzymes ^+^	5 (9)
Length of hospitalization (days) *	13 ± 6
Laboratory parameters: *	
- CRP (mg/L) (ref: <5)	154.1 ± 87.7
- PCT (ng/mL) (ref: <0.5)	9.17 ± 25.8
- Ferritin (times upper the normal value)	8.3 ± 8.8
- Ferritin (ng/mL)	832.2 ± 959.2
- AST (IU/L) (ref: <40)	50 ± 55
- ALT (IU/L) (ref: <40)	49 ± 70
- AMS (U/L) (ref: <80)	47 ± 48
- Lipase (U/L) (ref: <60)	38 ± 46

^+^ Number and percentage. * Mean and standard deviation. ALI: acute liver injury; CRP: C-reactive protein; PCT: procalcitonin; AST: aspartate aminotransferases; ALT: alanine aminotransferases; AMS: amylase.

**Table 2 children-09-00575-t002:** Baseline characteristics of patients with MIS-C with and without ALI.

Variables	ALI	Normal Transaminases	*p*
Number	26	29	
Males ^+^	9 (36)	20 (69)	0.02
Age in years *	8.3 ± 3.9	4.8 ± 2.7	0.0004
Obesity ^+^	4	4	ns
Duration of symptoms before hospital admission in days *	4.23 ± 2	3.72 ± 2.2	ns
Length of hospitalization in days *	13.9 ± 6.2	11.9 ± 5.9	ns
Inflammatory parameters *:			ns
- CRP (mg/L) (ref: <5)	154.1 ± 87.7	138.7 ± 68.7	
- PCT (ng/mL) (ref: <0.5)	13.7 ± 36.3	5.4 ± 8.4	
- Ferritin (times the ULN)	10.4 ± 7.3	6.4 ± 9.6	
- Ferritin (ng/mL)	916 ± 874.6	735.5 ± 1467.9	
- IL-6 (pg/mL) (ref: <5)	88.9 ± 83	200.5 ± 287	
- D-dimer (ng/mL) (ref: <270)	1995 ± 2988	1348 ± 1099	
- Fibrinogen (mg/dL) (ref: 180–400)	641.4 ± 229	560 ± 191	
- AMS (U/L) (ref: <80)	64 ± 63	32 ± 15	0.01
- Lipase (U/L) (ref: <60)	52 ± 55	19 ± 17	ns
Troponin >ULN ^+^	17 (65)	10 (34.5)	0.02
BNP * (pg/mL) (ref: <100)	524 ± 1014	417 ± 652	ns
Blood count:			ns
- Total leukocytes (count/µL)	10,895 ± 5878	12,231 ± 4582	
- Lymphocytes (count/µL)	1519 ± 1627	2346 ± 2348	
- Platelets (count/µL)	199,923 ± 99,444	231,241 ± 93,986	

^+^ Number and percentage. * Mean and standard deviation. ALI: acute liver injury; CRP: C-reactive protein; PCT: procalcitonin; AMS: amylase; ULN: upper level of normality.

**Table 3 children-09-00575-t003:** Baseline characteristics of patients with MIS-C with and without pancreatic involvement.

Variables	Raised Pancreatic Enzymes	Normal Pancreatic Enzymes	*p*
Number	24	31	
Males ^+^	10 (42%)	19 (61%)	ns
Age in years *	7.4 ± 3.5	5.7 ± 3.7	ns
Obesity ^+^	4	4	ns
Duration of symptoms before hospital admission (days) *	4.13 ± 2.3	11.6 ± 2	ns
Length of hospitalization (days) *	14.7 ± 5.6	11.6 ± 6.1	0.004
Inflammatory parameters: *			ns
- CRP (mg/L) (ref: <5)	172.2 ± 8.2	139.3 ± 76.6	
- PCT (ng/mL) (ref: <0.5)	7.2 ± 8.2	11 ± 33.7	
- Ferritin (times the ULN)	8.5 ± 6.8	8.2 ± 10.2	
- Ferritin (ng/mL)	886.8 ± 906.5	790.4 ± 1429.8	
- IL-6 (pg/mL) (ref: <5)	145 ± 284	153 ± 155	
- D-dimer (ng/mL) (ref: <270)	1932 ± 2816	1402 ± 1459	
- Fibrinogen (mg/dL) (ref: 180–400)	650.5 ± 241.8	553.1 ± 172.4	
- AST (IU/L) (ref: <40)	31 ± 12	43 ± 44	ns
- ALT (IU/L) (ref: <40)	47 ± 27	35 ± 32	
Tryglicerides (mg/dL)	203 ± 95	193 ± 159	ns
Blood count:			ns
- Total leukocytes (count/µL)	11,076 ± 4583.6	12,036 ± 5752	
- Lymphocytes (count/µL)	1910 ± 2482	1992 ± 1681	
- Platelets (count/µL)	222,120 ± 95,060	211,710 ± 94,439	

^+^ Number and percentage. * Mean and standard deviation. CRP: C-reactive protein; PCT: procalcitonin; AST: aspartate aminotransferases; ALT: alanine aminotransferases; AMS: amylase; ULN: upper level of normality.

## Data Availability

Not applicable.

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
