# Peer review of "Liver and Pancreatic Involvement in Children with Multisystem Inflammatory Syndrome Related to SARS-CoV-2: A Monocentric Study"

_children, 2022, doi:10.3390/children9040575_

Round 1

Reviewer 1 Report

Antonietta Giannattasio et al. present a monocentric study of the MIS-C cohort focusing on the liver and pancreatic involvement. A great advantage is the significant number of enrolled cases and the complete evaluation of the hepatic and pancreatic function. As a second aim, the authors evaluated the clinical, laboratory and imaging data and looked for correlations between acute liver injury, elevated pancreatic enzymes and other disease outcomes. 
The title is relevant and informative. 
The abstract represents the main results and conclusions. 
The introduction is appropriate in length and quality.
The study design, study subjects and methods descriptions are clearly described.
The demographic description of the cohort is described clearly and in detail.
Тhe two tables are informative.
The liver involvement is analyzed in detail - hepatocellular injury, markers of cholestasis, synthetic liver function, hemostasis.
The results are well and understandably described. 
The discussion is informative but focused predominantly on pancreatic involvement. 

What is the explanation for the progressively elevating aminotransferases in 10 cases (line 129) during hospitalization? Are those children with normal ASAT and ALAT at admission? Were they with a more severe clinical course? Did you find any correlations in this subgroup with the hypercoagulation state or pathological changes in the liver vascularization (v. portae, v. hepatica)?
How do you have a hypothesis explaining why elevated Troponin but not BNP correlate with the ALI?

Reviewer 2 Report

Thank you very much for the opportunity to read the manuscript entitled “Liver and pancreatic involvement in children with Multisystem Inflammatory Syndrome related to SARS-CoV-2: a monocentric study”. This paper provides an interesting overview of liver and pancreatic involvement in MIS-C patients, findings not usually described in these kind of patients. Introduction and methods are clear, data are presented and discussed appropriately, but see suggestions below that are encouraged to be implemented.

Manuscript has potential to be published, but changes are required.

Major changes:

  • What’s the reason for the different therapies administered (IVIG+/-steroids)? Any guidelines followed to therapy approach?
  • Include AST, ALT, amylase and lipase levels in all tables
  • Laboratory values shown at the tables are those presented at diagnosis, at admission or worse values during admission? Timing of measurements could help to understand these patients and in a potential creation of guidelines for MIS-C patients
  • It is indicated that in cases of hypertransaminasemia, other causes were excluded... Please, provide more details about the only patient with severe hypertransaminasemia (maybe other information redundant with data included in the tables can be removed if limit of words is reached)
  • Cases of steatosis observed, apparently 1 case was due to obesity, but the other 2? Were those 2 cases self-limited? Please, provide more information
  • Describe limitations of the study

Minor changes:

  • Include troponin and BNP levels in tables, rather than in the text, simplifying the text.
  • L108: South East Asia
  • L136-138: include this information in Table 2

Reviewer 3 Report

This study reviews the frequency of liver and pancreatic involvement in children with multisystem inflammatory syndrome associated to SARS-CoV-2, from a single center in Italy, and their association with other disease features and outcomes. The study is well written and addresses a topic that still remains poorly explored. However, some points need to be addressed to provide more clarity to the readers.

MAJOR COMMENTS:

  • The cohort designation as large sized is arbitrary, and in light of several previous studies including cohorts of >200 and even >500 patients, this cohort would be considered moderately sized at most. Please change this designation throughout the paper.

  • Why was a cutoff of 14 years of age chosen for inclusion? As older patients in the study were identified to have a higher frequency of liver injury, it would have been interesting to explore these data in patients older than 14 years as well.

  • Could the authors please clarify in the methods whether the study was prospective or retrospective? Based on the description it appears to have been a prospective design, but the phrase in lines 65-66 “only patients with … complete medical chart reviews were included” brings into question whether data were collected retrospectively.

  • In the methods, please clarify how often were labs collected and whether this was for study purposes or based on clinical need. Similarly, please clarify timing of follow up liver abdominal beyond discharge. Please also add how long after discharge were the patients followed and whether additional laboratory monitoring was conducted in the outpatient setting, as well as how long it took for patients with liver and pancreatic lab abnormalities to revert to normal values.

  • Please clarify why only amylase was chosen to define groups considered to have pancreatic injury. In the methods section, authors state that both amylase and lipase were accounted for in the definition but in the results the groups are defined based on amylase level only. Furthermore, lipase is a more specific marker of pancreatic injury – the authors should consider re defining the groups as high lipase vs normal lipase and leaving amylase as a secondary marker.

  • When considering associations of liver injury and inflammatory markers, why was a cutoff of 1000 chosen for ferritin level? The definition of MAS has a more conservative cutoff (~680) which would also give context to the findings if a predefined cutoff is used.

  • As pointed out by the authors, frequency of pancreatic injury in this cohort was low, with involvement being mild, and full recovery was observed in all patients. If not outcome-altering and not leading to changes in therapy, why should clinicians monitor pancreatic enzymes routinely, particularly if it led to unnecessarily longer stays in this cohort?

  • The authors should avoid overstatements throughout the paper. For example, in the last paragraph, the authors state that liver and pancreatic involvement are commonly observed in MIS-C patients – while this is true for their cohort, it is unclear if this applies to all patients with MIS-C. Please frame this type of conclusions to the study in question.

  • The authors should include a paragraph in the discussion were the limitations of the study including: smaller size compared to other cohorts, lack of multivariate analysis, and possibility that differences between the groups in terms of type of treatment and other parameters were not detected due to the study being underpowered.

Round 2

Reviewer 3 Report

Thank you for addressing the comments.